# Applying height differentiation of tactile symbols to reduce the minimum horizontal distances between them on tactile maps

Jakub Wabiński[1]*, Emilia Śmiechowska-Petrovskij[2], Albina Mościcka[1]

1 Institute of Geodesy, Faculty of Civil Engineering and Geodesy, Military University of Technology, Warsaw, Poland, 2 Department of Special Education, Faculty of Educational Sciences, Cardinal Stefan Wyszynski University, Warsaw, Poland

* Jakub.wabinski@wat.edu.pl

**Data Availability Statement:** The dataset in a form of Excel worksheet has been uploaded to Data Archiving and Networking Services (DANS). The DOI is https://doi.org/10.17026/dans-27r-sa9v.

## Abstract

In this paper, we wanted to verify the hypothesis that extruding cartographic symbols on tactile maps to different heights might allow reducing the minimum (suggested in the literature) horizontal distances between them, without impacting the overall map's legibility. This approach might allow preparing tactile maps in smaller scales and thus, reducing production cost, or putting additional spatial information on the same map sheet that would not fit otherwise. To verify the hypothesis we have prepared 6 different stimuli variants with or without height differentiation applied and different horizontal distances between tactile symbols adopted (1 mm, 2 mm and 3 mm). In the controlled study sessions with 30 participants with visual impairments we have measured the times required for solving 3 different spatial tasks on 3D printed tactile stimuli. We have also performed qualitative analysis to learn participants' opinions about the proposed design and materials used. It turns out that applying height differentiation not only results in shorter times required for solving spatial tasks but is also considered by blind individuals as a convenient improvement in terms of use comfort and allows reduction of recommended minimum horizontal distances between symbols on tactile maps.

## Introduction

A growing number of maps being produced is connected with the increasing demand for information provided in a very concise and convenient way, available for quick assimilation. These conditions, undoubtedly, are fulfilled by using cartographic form that shows spatial relations in a direct way, as a reflection of these relations in reality [1]. However, not many of the maps produced are of truly high quality. Besides, the vast majority of modern maps are digital, and thus, usually unavailable for people with special needs, such as people with visual impairment (PVI). This results in a situation, where access to high-quality tactile maps is incomparably more difficult than to their classic counterparts.

Consequently, students with visual impairments have difficulties with understanding of the maps. Most tactile maps are available in Braille textbooks or atlases, but are prepared using

**Funding:** This research was funded by Military University of Technology, Faculty of Civil Engineering and Geodesy, grant number 531-4000-22-871/UGB/2021.

**Competing interests:** The authors have declared that no competing interests exist.

inappropriate techniques, e.g. Tiger embosser that is not efficient in differentiating tactile symbols [2]. Too high complexity of tactile graphics is the main factor lowering their comprehensibility by PVI (e.g. adaptation of too detailed maps without proper generalization). Thus, preparing them while using a production technique that increases the sense of complexity and is tactilely unfriendly is an inefficient strategy. As a result, almost 47% of blind students are unable to follow their peers with normal vision when solving tasks in a class [3].

Other than that, it turns out that students who learned a given area by scanning tactile maps were considerably more proficient in unguided route following than those who based on direct experience or verbal instructions [4].

But why are not digital maps converted into tactile form, legible by PVI? Tactile map development is complicated and expensive. They often require personalization–either due to the needs of a reader or because they are supposed to map specific areas or even routes. Tactile maps also require a decent level of generalization because of the way PVI perceive maps— using their sense of touch or damaged sight. Not only is the resolution of a finger approximately 10 times worse than that of an eye (in normal conditions), but also tactile maps are usually being read fragment by fragment, out of which an image of the whole map is built up in a reader's memory [5]. This makes the production of tactile maps complicated and difficult to automate. A number of solutions for automatic tactile map generation have been proposed but none of them can be treated as a holistic solution that meets all the requirements of PVI [6]. Moreover, many of the commonly used production methods (e.g. thermoforming or relief screen printing) are cost-effective only in the case of mass production, whereas tactile maps usually have to be printed in small quantities.

The amount of information conveyed by each map depends on the number and type of elements used on it, as well as on cartographic symbols used for presentation [7]. Due to the need of strong generalization of map elements that includes sparse arrangement of tactile symbols, allowing PVI to read them, it is not possible to fit the same number of cartographic symbols on a tactile map as on its classic counterpart. It is important to develop new solutions that could allow compacting tactile map content or increasing the amount of information of single tactile symbols. Unfortunately, both are hard to achieve due to the perceptive limitations of PVI.

When creating tactile equivalents of classic maps, a number of solutions can be used to convey the same amount of information. Tactile counterparts can be for example prepared in larger formats. But limitations in terms of maximum dimensions of a tactile map sheet do exist–a seated reader has to cover the whole sheet with his/her arms [8]. Another possibility is to create a series of maps of the same area, each of them covering only a part of the topic [9]. This solution, however, is more expensive and time consuming. An alternative is to try to increase the information value of a particular tactile map or its legibility. On classic maps this can be achieved by using various graphic variables that facilitate distinguishing cartographic symbols. Unfortunately, when designing maps to be read using touch, cartographers cannot use variables based on colour–the ones that are the easiest to perceive [10]. Instead, they are limited to a set of haptic variables [11] (Fig 1).

Most haptic variables are already widely used on tactile maps and it is difficult to find their further differentiation possibilities. The variable that seems to have potential in this matter is height. This is because the hitherto used methods of tactile maps printing make it possible to print symbols of the varying height only to some limited extent. But even though it is possible, not many tactile maps offer height differentiation of cartographic symbols. The emerging new production techniques, such as 3D printing, make it possible to print almost any symbols and thus, to freely differentiate their heights.

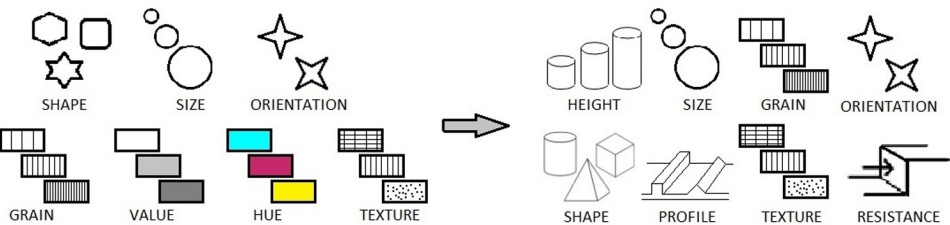

**Fig 1. Graphic and haptic variables, source: Based on [11].**

## Information conveyed by tactile maps

The information value of a map is influenced not only by the number of content elements present on a given map fragment (e.g. 1 cm$^2$), but also the amount of information transmitted by individual symbols and the structure of their arrangement on a given map fragment [12, 13]. New tactile map design principles together with the new editorial rules should be developed to increase tactile maps' information values, while still keeping them legible. We have already proven that introducing height differentiation of symbols on a tactile map leads to the increase of information value [11].

The importance of this concept has been widely discussed in the literature. One of the methods to partly eliminate clutter on tactile maps is to use height differentiation of tactile symbols [14, 15]. Maps with different cartographic symbols, varying in heights from the background material, are easier to read and understand [9]. A more recent study showed a strong user preference towards 3D plans in favour of 2.5D tactile graphics [16].

## Design requirements for tactile maps

Tactile maps design guidelines in many countries suggest minimum distances to be kept between tactile symbols in order to maintain map's legibility. The most commonly quoted value in the literature on this subject is 3 mm [17–19], which is related with an average resolution of a finger, estimated to be between 2.4 and 3 mm [20]. This distance is sufficient in case of two highly contrasting symbols. In order to distinguish two symbols placed next to each other that are similar in shape or smaller in size than suggested in literature, this distance should be increased to 5–6 mm [17].

Taking the above into consideration, we would like to analyse the impact of tactile symbols height differentiation and related new editorial proposals on the amount of information conveyed by tactile maps. Therefore, **the aim of this study was to examine whether the height differentiation of tactile symbols enables reduction of horizontal distances between them, while maintaining legibility of a tactile map**. We assumed that applying height differentiation improves the legibility of maps. We defined the hypothesis that height differentiation of tactile symbols also allows the reduction of suggested in literature distances between particular symbols. **We would like to determine to what extent these distances can be reduced, if at all.** Our hypothesis was that height differentiation of symbols on tactile maps may either allow an increase of the number of cartographic symbols used (thereby, real-world elements presented) on a single map, while still keeping it legible or, to a decrease of map scale as well as map sheet dimensions and thus, the related expenses. We also assumed that in such case less generalization could be applied and thanks to that, real-world objects could be represented on tactile maps more realistically and with less distortions. This is especially important in case of elements, whose location and/or size have to be precisely rendered. Such solutions will finally increase the information value of a particular map.

Therefore, the following research questions were defined:

**RQ1:** Does height differentiation of cartographic symbols on tactile maps allow reduction of recommended minimum distances between them?

**RQ2:** To what extent can minimum horizontal distances between particular symbols be reduced for symbols with varying heights?

**RQ3:** Do height differentiation of cartographic symbols and reduction of minimum distances between them increase information value of tactile maps?

Our findings can lead to more efficient tactile map production and facilitate the process of their development, including automatic generalization. As a consequence, this may contribute to reduction of the costs of tactile maps production.

## Materials and methods

We have divided our research into three stages: (1) developing tactile stimuli, (2) human subject testing along with statistical analysis of the results, and (3) information value evaluation. Each of these stages is described in the following subsections.

### Developing tactile stimuli

To verify our hypothesis we had to design a set of tactile stimuli that would mimic tactile maps. These *pseudomaps* [as referred in 19] were not based on any particular spatial data but were purposely designed for this research. Their design was supposed to allow performing simple spatial tasks evaluating the potential of tactile symbols' height differentiation. For this purpose we have prepared a number of variants of the same tactile stimulus. We have chosen 3 levels of complexity, defined by the minimum horizontal distances between symbols to be kept. For each level of complexity, we have designed 2 stimulus versions: with and without height differentiation.

We used the versions without height differentiation (D1, D2, D3 in Fig 2) to examine the effect of the minimum horizontal distances reduction on the map readability. These stimuli variants also constituted a reference data for examining the impact of height differentiation of

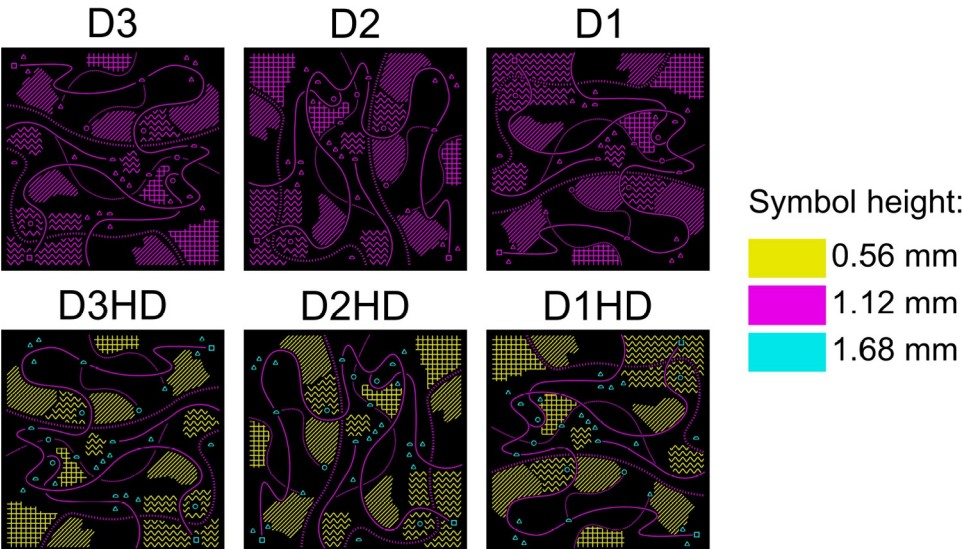

**Fig 2. The stimuli used in our study.** Different colours indicate heights of particular symbol types. Mirroring and rotations are visible.

symbols on the map readability–to find answer to the RQ1. We used the variants with height differentiation applied (D1HD, D2HD, D3HD in Fig 2) to test to what extent the spacing between the symbols could be reduced, providing answer to the RQ2.

In order to minimize potential confounding variables, we have prepared 7 different versions of alignment of the symbols on particular stimuli, so that none of the variants has the same arrangement of symbols. By applying rotations and mirror reflections we wanted to avoid participants' memorization of specific patterns on the stimuli, while repeating tasks. We presented the stimuli variants to the participants in random order.

We have maintained the same number of symbols on each stimuli that allowed us to eliminate potential confounding variables in the form of modified number or character of tactile symbols used. We have also prepared an extra "pilot" stimuli that was presented to participants first so that they could get familiar with the maps' content and their legend. This additional stimuli version was not used in the performance analysis. The universal legend was printed separately. We have glued the 3D printed symbol list to a swell-paper sheet with Braille descriptions.

As it was confirmed in one of our previous studies [21], the minimum distances of 3 mm between contrasting tactile symbols are sufficient for 3D printed tactile maps to be read correctly. In this research we wanted to verify if it is possible to further reduce these distances, while still being able to solve spatial tasks. The proposed stimuli variants were prepared as described in Table 1. Each of the stimuli variants has its own stimulus code, e.g. D3HD stands for 3 mm minimum horizontal distances between symbols with height differentiation applied, whereas for D1 variant 1 mm of minimum horizontal distances were applied with all symbols put at the same height. Visual representation of the stimuli variants used in our study is presented in Fig 2.

Applying mirror reflections causes some of the area symbol textures (in orange–Fig 3) to change their orientation. We have modified these area symbols to keep their orientation the same across every stimuli but without altering their outlines that were used for information value calculations. This operation had no impact on symbols outlines but resulted in slightly different numbers of elements and their arrangement within outlines.

In future tasks related with production of tactile maps, the reduction of minimum distances between particular tactile symbols could allow placing more symbols of any type on the same map sheet, or to more realistically depict the real-world objects (e.g. their borders or outlines). In this research, the number of symbols remained the same across all the stimuli, in order not to introduce potential confounding variables. Instead of introducing new tactile symbols, we used the additional space gained after reducing the minimum distances and filled them with symbols already existent on stimuli (Fig 4).

**Table 1. Tactile stimuli variants.**

| Stimulus code | Complexity level: minimum distances | Height differentiation [mm] | Rotation | Mirror reflection |
|---|---|---|---|---|
| D3 | Low: 3 mm | No (1.12) | 0˚ | Yes |
| D2 | Medium: 2 mm | No (1.12) | 270˚ | No |
| D1 | High: 1 mm | No (1.12) | 180˚ | No |
| D3HD | Low: 3 mm | Yes (0.56, 1.12, 1.68) | 0˚ | No |
| D2HD | Medium: 2 mm | Yes (0.56, 1.12, 1.68) | 270˚ | Yes |
| D1HD | High: 1 mm | Yes (0.56, 1.12, 1.68) | 180˚ | Yes |
| Pilot [a] | varies | Yes (varied) | 90 | No |

[a] not considered in the performance analysis

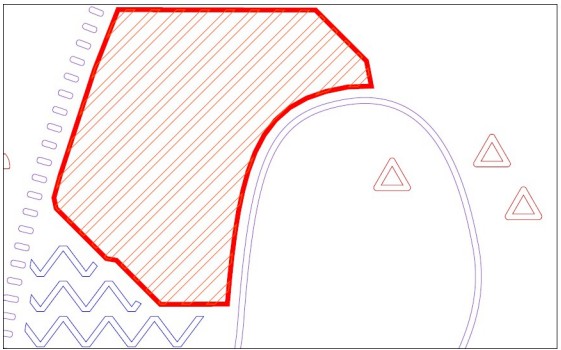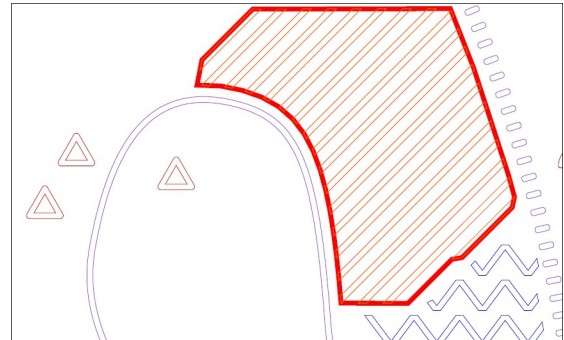

**Fig 3. The impact of mirror reflection on area symbol texture.**

The knowledge and good practices regarding the design of symbols on classic maps cannot be directly applied as a guide for production of tactile symbols [22]. Even though the standardization of tactile symbols has not yet been fully achieved [23], some guidelines and good case examples on how to design tactile maps and symbols on them do exist [e.g. 17, 18]. Based on these guidelines and our past experience with 3D printed tactile maps [11] we have carefully selected tactile symbols to be used on the stimuli. Our aim was to select the symbols that are commonly used on tactile maps, especially the 3D printed ones (cf. Table 2). We have chosen easily distinguishable symbols that were used together on particular map sheets in past research and are well known to PVI. By doing this, we wanted to remove potential confounding variables and evaluate only the impact of height differentiation and minimum distances between particular symbols.

The extrusion heights of particular symbols categories are based on past experiments [14, 21]. The larger the symbol, the less extrusion it requires. Thus, the biggest extrusion has been applied to point symbols (1.68 mm). Line symbols have been extruded to 1.12 mm height, whereas area symbols are 0.56 mm in height. On stimuli variants with no height differentiation applied, a fixed height of 1.12 mm was used for every tactile symbol. These values are a multiple of selected layer thickness of 3D printing process (0.14 mm).

According to the report of the National Federation of the Blind [32], only 10% of 1.3 million legally blind people in the United States are Braille readers. Today, children with visual impairments express strong preference for audio materials instead of Braille publications–Braille on paper is declining in Europe [33]. For this reason we resigned from placing Braille labels on

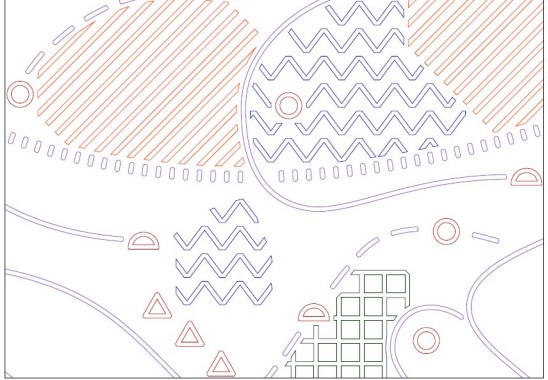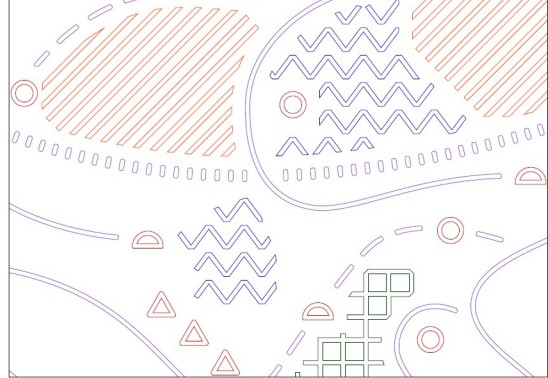

**Fig 4. The increase of free space that can be filled with tactile symbols when reducing minimum distances between symbols.**

**Table 2. The symbols used in the research.**

| POINT SYMBOLS | | | | |
|---|---|---|---|---|
| *CONTOUR WIDTH: 1 MM, EXTRUSION HEIGHT: 1.68 MM* | | | | |
| **SYMBOL GEOMETRY** | **Dimensions [mm]** | **Referenced in** | **Purpose of use** | **Accompanying symbols** |
| **TRIANGLE** | length: 6.25 | [24] | n/a | zig-zag, check pattern, line pattern, solid line |
| | width: 5.5 | [25] | point of interest | circle, square, solid line |
| **CIRCLE** | diameter: 5.75 | [9][a] | junction of paths | solid line, dashed line |
| | | [26] | telephone booth | line pattern |
| | | [25] | point of interest | triangle, square, solid line |
| **SQUARE** | length: 5.75 | [27][b] | telephone booth | solid line, dashed line, pattern line |
| | | [25] | point of interest | triangle, circle, solid line |
| **HALF ELLIPSE** | length: 6.75 | [21] | lake | solid line, zig-zag, line pattern |
| | width: 3.75 | [28][c] | lake | circle, solid line, zig-zag, line pattern |
| **SYMBOL GEOMETRY** | **Dimensions [mm]** | **Referenced in** | **Purpose of use** | **Accompanying symbols** |
| **SOLID LINE** | line width: 1 | [24] | n/a | triangle, zig-zag, check pattern, dashed line |
| | | [27][b] | main road | square, dashed line |
| | | [21] | river | half ellipse, zig-zag, line pattern |
| **DASHED LINE** | line width: 0.5 | [27][b] | railway | square, solid line |
| | dash thickness: 0.5 | | | |
| | dash width: 1.5 | [24] | n/a | triangle, zig-zag, check pattern, solid line |
| | gap: 1.5 | | | |
| **PATTERN LINE** | dash thickness: 1 | [29][a] | fence | circle, solid line, dashed line |
| | dash width: 2.5 | | | |
| | gap: 2 | [30] | tramway | solid line |
| **SYMBOL GEOMETRY** | **Dimensions [mm]** | **Referenced in** | **Purpose of use** | **Accompanying symbols** |
| **ZIG-ZAG** | line width: 1 | [17][a] | recommended texture | can be used with any textures (universal) |
| | vert. dist.: 6 | [31] | n/a | triangle, check pattern, solid line, dashed line |
| | horiz. dist.: 7.75 | [21] | mountains | half ellipse, solid line, zig-zag, line pattern |
| **CHECK PATTERN** | line width: 1 vertical dist.: 5 | [17][a] | recommended texture | can be used with any textures (universal) |
| | horizontal dist.: 5 | [31] | n/a | triangle, solid line, dashed line, zig-zag, line pattern |
| **LINE PATTERN** | line width: 1 | [26] | green area | circle |
| | vert. dist.: 4.25 | | | |
| | horiz. dist.: 4.25 | [21] | highlands | half ellipse, solid line, zig-zag |
| | angle: 45˚/225˚ | [17][a] | recommended texture | can be used with any textures (universal) |

Other than 3D printing

[a] Microcapsule method (swell paper)

[b] Manual methods

[c] Thermoforming method

the stimuli. We did not want to limit study participants only to those who can read Braille. Braille descriptions were only placed on a map legend but the researchers offered their help in case participants were not able to read the descriptions.

We have decided to use a 3D printing method for stimuli production, and more specifically–Fused Deposition Modelling (FDM). This method uses thermoplastic material to form a physical model layer by layer on the 3D printer's surface. We have chosen this production

method due to its characteristics and applicability in tactile aids production [e.g. 34–36]. 3D printing is perfect for rapid prototyping and thus, new map designs can be verified at fast pace and at low cost.

## Human subject testing

In order to verify our hypothesis, we have planned a research activity that involved human testing of PVI both congenitally/early (under 5 years old) and adventitiously blind. Participants were asked to solve a number of basic location tasks using tactile map stimuli, designed for this particular research. Every participant had to repeat three location tasks on each stimulus variant (18 tasks total). Every task was related with different geometry type of cartographic symbols:

- Task 1—locate 5 point symbols on the map of a given type (the reference symbol to be found was shown to participants before the actual task on map legend).

- Task 2—locate 5 area symbols in the same manner.

- Task 3—follow the path (line symbol) to the specific point from their current location (indicated at the start by a researcher), while avoiding obstacles (both in a form of area and point symbols).

By asking participants to solve similar spatial tasks on different variants of the same tactile stimuli, we have verified the impact of different approaches of height differentiation and placement of tactile symbols on their overall performance. Each participant, prior to performing the designed tasks, was allowed to examine the pilot map (stimulus) and the associated legend briefly–explore its dimensions and material used as well as the symbols used.

The recruitment process for the study was carried out by the authors with help of the Polish Blind Association (https://pzn.org.pl/). People willing to take part in our research study were asked to indicate their interest through a web-based response form. The recruitment forms were prepared in accordance with the Research Protocol (S1 File). Before conducting the actual recruitment phase, we have tested the research procedure in a pilot study with 2 PVI.

Every participant was examined individually. We were documenting their performance of solving the tasks (time required) as well as their behaviour during that process by recording their body movement on video. After sessions, participants were asked to answer a number of questions to get participants' feedback (qualitative analysis). Each participant took part in one session lasting approximately 60 minutes. The tests were conducted according to the previously established schedule described in the Research Protocol.

The performance results and answers from the questionnaires were later anonymously aggregated and analysed using statistical measures.

To confirm our hypothesis and answer RQ1 and RQ2 all variants of the stimuli were necessary. Stimuli with all the symbols extruded to the same height served as the reference for stimuli with height differentiation applied. Comparison of stimuli with and without height differentiation was a key point to determine what kind of changes we obtain thanks to this design modification (RQ1). Comparison of stimuli with height differentiation was necessary to measure the impact of these changes (RQ2). Therefore, all the performance results were included in the statistical analyses.

We have used non-parametric versions of all the statistical tests as our data do not meet the assumptions of parametric tests, such as normal sampling distribution (result of Kolmogorov-Smirnov test was of statistical significance, $p < 0.05$). To analyse our data, we have conducted an overall Friedman rank sum test (for repeated measures of our respondents solving task in

six stimuli variants), Kruskal-Wallis test (independent samples for two or more groups) and the Mann-Whitney U test (for two independent groups) to see if there were any significant differences in the measured variables among the various conditions. For the Friedman rank sum test, we performed a pairwise Wilcoxon signed rank test as the post hoc test, with a Benjamini-Hochberg (BF) correction. We followed the Kruskal-Wallis tests with a Dunn's post hoc test along with Benjamini-Hochberg correction, where we compared specific pairs. Data were analysed using IBM Statistics 27.

### Information value evaluation

We have also examined the influence of cartographic symbols' height differentiation and their arrangement on the overall information value of the tactile stimuli prepared. One of measures that defines the informational potential of a map is structural measure of information, first proposed by Salistchev [7]. The structural measure of information was used for evaluation of tactile maps in one of our past research [11]. Due to the fact that in this research only simple tactile stimuli were used (rather than regular tactile maps), we had to modify the original formula for measuring structural measure of information. The weights applied to particular symbol types are the same—each symbol defines only location and real-world object category:

$$SMI = \sum_{i=1}^{k} a_i + \sum_{i=1}^{l} b_i + \sum_{i=1}^{m} c_i \qquad (1)$$

where $k$, $l$, $m$ is the number of point, line and area cartographic symbols of a given type, $a_i$ is the number of point symbols of particular type, $b_i$ and $c_i$ are respectively: the lengths of particular types of linear symbols and boundary lines of area symbols.

It is not possible to compare the measurements for particular symbol types directly, due to significant differences in the measured values between different geometries. Point symbols are counted one by one, whereas for line and area symbols, geometrical measures are used. Using normalization in that case was impossible, as the number of point symbols on each stimulus is the same. For this reason we have decided to examine only the geometrical measures of line and area symbols to determine the potential increase of information value, when reducing the minimal distances between tactile symbols.

### Results

The resulting stimuli are 22 by 22 cm in planar size (Fig 5). We used real-scale vector drawings for the geometrical measurements that determine the potential increase of information value.

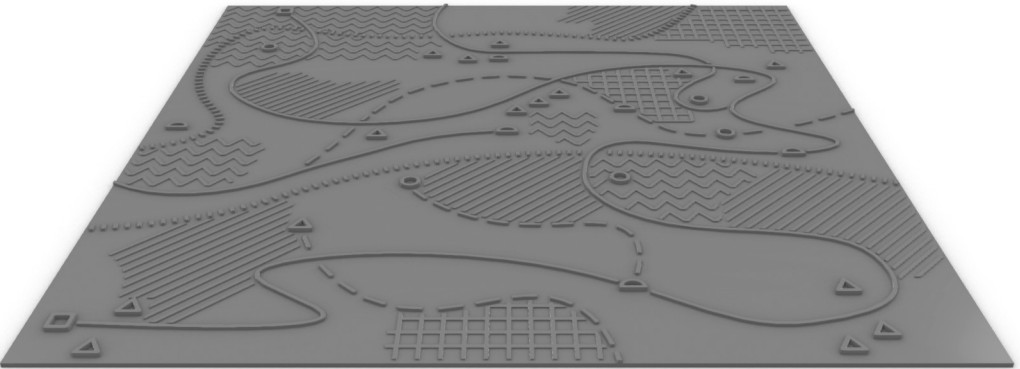

**Fig 5. Digital 3D Model of one of the stimuli variants.**

Two exemplary vector drawings of the stimuli with 3 mm offsets around every symbol are presented in Fig 6. In this case a mirror reflection has been used for differentiation and to prevent the map content from being memorized.

The stimulus base thickness was set to 1.2 mm. We have proven during the pilot study that this thickness provides sufficient durability of the material. We have prepared 7 physical variants of the 3D stimuli, using PLA material. In order to provide an element for stimuli orientation, specially designed convex triangles were 3D printed and then glued to the upper right corner of each stimulus.

## Testing phase

The testing phase was conducted at the Polish Blind Association building in Warsaw, Poland. It lasted for 3 days in May 2021. Out of over 50 applicants, we have selected a group of 30 people (aged 16–65) that took part in our study. The number of participants examined allowed us to avoid potential confounds related to the repetition of location tasks on a number of variants of the same map by a single reader (Fig 7).

Based on the application forms, we have tried to choose the most representative group (statistically diverse). We recruited 16 male and 14 female participants, the mean age was 38.9 (SD = 10.45). Most of them (22) were congenitally blind or lost their sight before 5 years old (early blind), whereas 7 persons had adventitious blindness (1 participant did not provide the moment of sight loss). The majority had high (12 individuals) or average experience (16) of using tactile maps according to their subjective assessment and high (23) or average experience (7) with Braille reading.

No statistically significant differences were noted between level of experience with tactile maps and Braille reading (Mann-Whitney U-test $\chi^2(3) = 1.815$, p = 0.612). Early blind participants had higher level of experience with Braille reading than adventitiously blind ($\chi^2(1) = 10.881$, p<0.05, mean rank value for early blind: 17.18; mean rank value for adventitiously blind: 8.14), but there was no statistically significant difference in level of experience with tactile maps between early blind and adventitiously blind participants ($\chi^2(1) = 0.270$, p = 0.603). The characteristics of the participants of our study are presented in Fig 8.

## Performance results

The main goal of the testing phase was to evaluate times required for solving the 3 spatial tasks. We assumed that the lower the average times needed to solve spatial tasks by the study participants, the more legible the particular tactile stimuli are. This means that the tactile stimuli

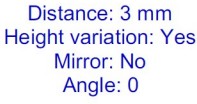
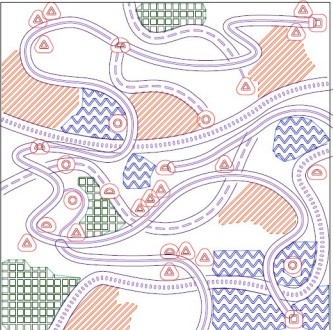
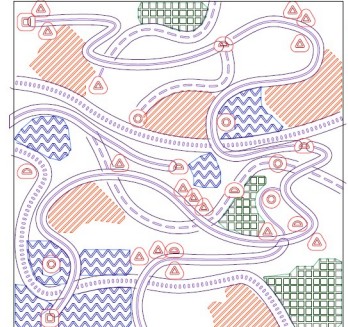

**Fig 6. A pair of vector drawings used for geometric calculations.** The applied 3 mm offsets are visible.

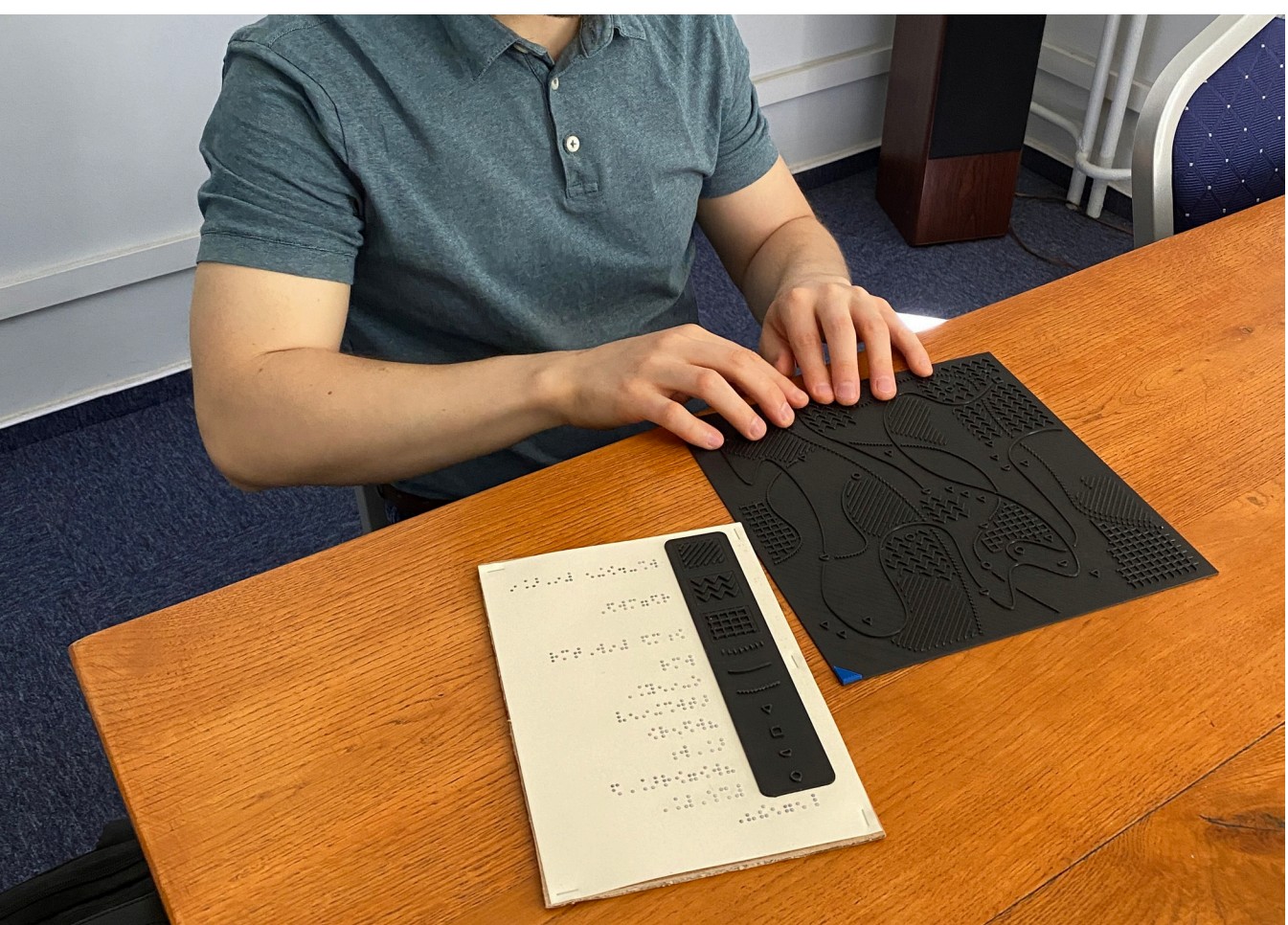

**Fig 7. One of the study participants during the testing phase, source: Own photo.**

variants with the lowest mean solving times and highest solved tasks rate are considered the best cartographic design options.

The maximum time available for solving a task was set to 5 minutes. If, after this time, the task remained unsolved, we informed the participants about this fact and moved onto the next tasks and/or stimulus variant. To determine average solving times of particular tasks, the unsolved ones were excluded to count real average solving time. Based on our observations, the first stimulus presented was usually characterized by slightly longer solving times than average for that stimulus variant but we avoided its impact by presenting particular variants to the study participants in random order.

The results of Mann-Whitney U test showed no significant differences in the average solving times for each stimulus variant and each task among congenitally/early and adventitiously blind study participants. Moreover, Kruskal-Wallis test showed no statistically significant differences between participants' level of experience with tactile maps or Braille reading and the average tasks solving times.

When it comes to finding point symbols (small elements) on tactile maps (task 1), the lack of height differentiation of the symbols directly affects the readability of the map (Tables 3–5). Decreasing the minimum distances between symbols to 1 mm made identification of the symbols impossible (93.3% of study participants). Reducing this distance to 2 mm gave a positive

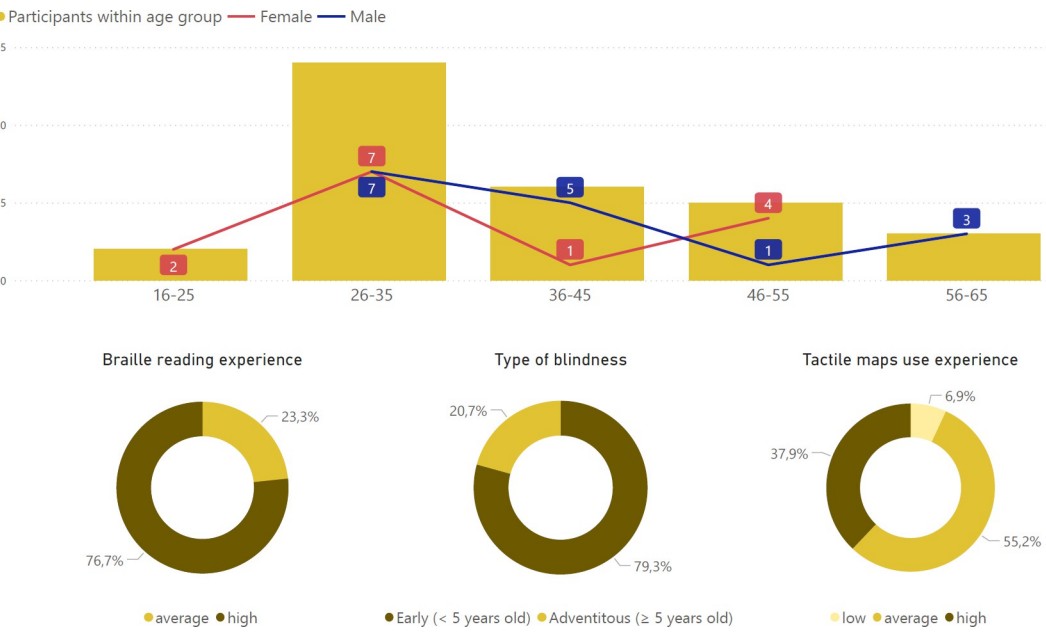

**Fig 8. Study group statistics.**

result only in 36.7% of cases, and to 3 mm—in 56.7% of cases. However, with height differentiation applied, the study participants were able to find and identify point symbols in 96.7% of cases, regardless of the distance between the elements. Table 3 shows the average solving times for each stimulus variant, including only those cases which were successfully completed. In the analyses of significance of differences, the D1 variant was excluded because the number of valid observations was less than 5 (2 cases).

The Friedman test result showed statistically significant differences between solving times of stimuli variants ($\chi^2(4) = 13.400$; $p < 0.05$). Pairwise Wilcoxon Signed-Rank tests with a BH correction let us detect that no statistically significant differences were found between the time of correct identification of the symbols and the minimal distances between them for stimuli variants with height differentiation applied (Tables 4 and 5). These variants are feasible and we can observe lower average solving times in comparison to variants without height differentiation.

For task 1, in the case of using the same distances between symbols, but above 1 mm, the height differentiation significantly impacts the time of symbol identification. For D2HD-D2

**Table 3. Task 1 statistics.**

| STIMULUS CODE | | D1 | D2 | D3 | D1HD | D2HD | D3HD |
|---|---|---|---|---|---|---|---|
| N | Valid | 2 | 11 | 17 | 29 | 29 | 29 |
| | Missing | 28 | 19 | 13 | 1 | 1 | 1 |
| MEAN | | 02:45 | 03:12 | 02:41 | 01:43 | 01:40 | 02:07 |
| MODE | | 02:15 | 04:59 | 00:43[a] | 01:15[a] | 01:28[a] | 02:15 |
| STD. DEVIATION | | 00:42 | 01:17 | 01:29 | 01:01 | 01:05 | 01:16 |
| MINIMUM | | 02:15 | 01:18 | 00:43 | 00:31 | 00:32 | 00:33 |
| MAXIMUM | | 03:15 | 04:59 | 04:56 | 04:38 | 05:00 | 04:59 |

[a] Multiple modes exist. The lowest value is shown

**Table 4. Wilcoxon signed ranks test—task 1.**

| STIMULI PAIR (CODES) | D3-D2 | D1HD-D2 | D2HD-D2 | D3HD-D2 | D1HD-D3 |
|---|---|---|---|---|---|
| Z | -1.540[a] | -2.401[a] | -2.936[a] | -2.312[a] | -1.888[a] |
| ASYMP. SIG. (2-TAILED) | .123 | .016 | .003 | .021 | .059 |
| STIMULI PAIR (CODES) | D2HD-D3 | D3HD-D3 | D2HD-D1HD | D3HD-D1HD | D3HD-D2HD |
| Z | -1.939[a] | -2.017[a] | -.384[a] | -.854[b] | -1.754[b] |
| ASYMP. SIG. (2-TAILED) | .052 | .044 | .701 | .393 | .079 |

[a] based on positive ranks

[b] based on negative ranks

**Table 5. Task 1 –ranks.**

| STIMULI PAIR (CODES) | | N | MEAN RANK | SUM OF RANKS |
|---|---|---|---|---|
| D3-D2 | Negative Ranks | 7[a] | 4.14 | 29.00 |
| | Positive Ranks | 1[b] | 7.00 | 7.00 |
| | Ties | 0[c] | | |
| | Total | 8 | | |
| D1HD-D2 | Negative Ranks | 10[d] | 6.00 | 60.00 |
| | Positive Ranks | 1[e] | 6.00 | 6.00 |
| | Ties | 0[f] | | |
| | Total | 11 | | |
| D2HD-D2 | Negative Ranks | 11[g] | 6.00 | 66.00 |
| | Positive Ranks | 0[h] | .00 | .00 |
| | Ties | 0[i] | | |
| | Total | 11 | | |
| D3HD-D2 | Negative Ranks | 10[j] | 5.90 | 59.00 |
| | Positive Ranks | 1[k] | 7.00 | 7.00 |
| | Ties | 0[l] | | |
| | Total | 11 | | |
| D1HD-D3 | Negative Ranks | 11[m] | 9.50 | 104.50 |
| | Positive Ranks | 5[n] | 6.30 | 31.50 |
| | Ties | 1[o] | | |
| | Total | 17 | | |
| D2HD-D3 | Negative Ranks | 11[p] | 9.59 | 105.50 |
| | Positive Ranks | 5[q] | 6.10 | 30.50 |
| | Ties | 0[r] | | |
| | Total | 16 | | |
| D3HD-D3 | Negative Ranks | 13[s] | 8.23 | 107.00 |
| | Positive Ranks | 3[t] | 9.67 | 29.00 |
| | Ties | 1[u] | | |
| | Total | 17 | | |
| D2HD-D1HD | Negative Ranks | 13[v] | 15.77 | 205.00 |
| | Positive Ranks | 14[w] | 12.36 | 173.00 |
| | Ties | 0[x] | | |
| | Total | 27 | | |

*(Continued)*

**Table 5.** (Continued)

| STIMULI PAIR (CODES) | | N | MEAN RANK | SUM OF RANKS |
|---|---|---|---|---|
| **D3HD-D1HD** | Negative Ranks | 13[y] | 12.73 | 165.50 |
| | Positive Ranks | 15[z] | 16.03 | 240.50 |
| | Ties | 0[aa] | | |
| | Total | 28 | | |
| **D3HD-D2HD** | Negative Ranks | 10[ab] | 11.60 | 116.00 |
| | Positive Ranks | 17[ac] | 15.41 | 262.00 |
| | Ties | 0[ad] | | |
| | Total | 27 | | |

**a.** D3 < D2

**b.** D3 > D2

**c.** D3 = D2

**d.** D1HD < D2

**e.** D1HD > D2

**f.** D1HD = D2

**g.** D2HD < D2

**h.** D2HD > D2

**i.** D2HD = D2

**j.** D3HD < D2

**k.** D3HD > D2

**l.** D3HD = D2

**m.** D1HD < D3

**n.** D1HD > D3

**o.** D1HD = D3

**p.** D2HD < D3

**q.** D2HD > D3

**r.** D2HD = D3

**s.** D3HD < D3

**t.** D3HD > D3

**u.** D3HD = D3

**v.** D2HD < D1HD

**w.** D2HD > D1HD

**x.** D2HD = D1HD

**y.** D3HD < D1HD

**z.** D3HD > D1HD

**aa.** D3HD = D1HD

**ab.** D3HD < D2HD

**ac.** D3HD > D2HD

**ad.** D3HD = D2HD

stimuli pair (Z = 2.017; p<0.05), D2HD average solving time was faster by 1:32 than D2. In case of D3HD-D3 (Z = 2.017; p <0.05) by 0:34. The study has also demonstrated other significant differences between variants. For the stimuli pair D1HD-D2 (Z = 2.401; p<0.05), D1HD average solving time was faster by 1:28 than D2. In the case of D3HD-D2 pair (Z = 2.312; p<0.05), by 1:05. Comparison of solving times for D3, D2HD and D1HD showed trend towards significance: D2HD average solving time was lower by 01:01 than D3 (Z = 1.888; p = 0.059) and D1HD on average lower by 0:58 than D3 (Z = 1.939; p = 0.052).

**Table 6. Task 2 statistics.**

| | STIMULUS CODE | D1 | D2 | D3 | D1HD | D2HD | D3HD |
|---|---|---|---|---|---|---|---|
| N | Valid | 30 | 30 | 30 | 30 | 30 | 30 |
| | Missing | 0 | 0 | 0 | 0 | 0 | 0 |
| MEAN | | 00:27 | 00:21 | 00:22 | 00:28 | 00:22 | 00:24 |
| MODE | | 00:20 | 00:14[a] | 00:16 | 00:12[a] | 00:10 | 00:16[a] |
| STD. DEVIATION | | 00:18 | 00:17 | 00:11 | 00:36 | 00:15 | 00:12 |
| MINIMUM | | 00:04 | 00:06 | 00:05 | 00:07 | 00:09 | 00:05 |
| MAXIMUM | | 01:19 | 01:25 | 00:53 | 03:16 | 01:13 | 01:01 |

[a] Multiple modes exist. The lowest value is shown

In conclusion, in case of point symbols, using height differentiation, while reducing distances between symbols, improves the identification process of point symbols in relation to the variants with larger distances between the symbols without height differentiation.

The second task that required identification of large area symbols was fairly easy for all the participants and the average solving times were almost the same across all the stimuli variants that we analysed (21–28 seconds: Table 6). The differences between times of identification of area symbols in stimuli variants are not statistically significant (Friedman's test $\chi^2(5) = 7.294$; p> 0.05). The times required for solving this task depended more on the scanning technique used than the way the particular stimulus was designed.

The third spatial task required the readers to track a path from starting point to the goal. To make things harder, the goal was hidden in the heights (zig-zag pattern). Thus, we can observe the impact of both increased minimum distances and height differentiation applied on the average results. For most participants (76.6%), solving this task on the D1 stimulus variant was impossible (cf. Tables 7–9). For all other samples we can observe slightly lower average solving times for stimuli variants with height differentiation applied (D1HD, D2HD, D3HD). The differences in the times of identification of line symbols were statistically significant (Friedman's test $\chi^2(4) = 18{,}554$; p<0.05).

According to the results of pairwise Wilcoxon Signed-Rank tests with a BH correction (Tables 8 and 9), tracking the path (line symbols) was the most effective when participants used stimuli variants with height differentiation applied (average solving time for D3HD: 00:29). Moreover, there were significant differences between average solving times for D3HD in comparison with D2HD and D1HD: 00:12 (Z = 3.557; p<0.05) and 00:41 (Z = 3.929; p<0.05) respectively. Differences between D1HD and D2HD in solving times were not statistically significant.

**Table 7. Task 3 statistics.**

| STIMULUS CODE | | D1 | D2 | D3 | D1HD | D2HD | D3HD |
|---|---|---|---|---|---|---|---|
| N | Valid | 7 | 30 | 30 | 30 | 30 | 30 |
| | Missing | 23 | 0 | 0 | 0 | 0 | 0 |
| MEAN | | 02:40 | 01:04 | 00:47 | 00:42 | 00:41 | 00:29 |
| MODE | | 00:55[a] | 00:29 | 00:30 | 00:19 | 00:23 | 00:15 |
| STD. DEVIATION | | 01:24 | 01:10 | 00:48 | 00:28 | 00:35 | 00:19 |
| MINIMUM | | 00:55 | 00:20 | 00:15 | 00:14 | 00:10 | 00:13 |
| MAXIMUM | | 05:00 | 04:52 | 03:45 | 02:14 | 02:34 | 01:26 |

[a] Multiple modes exist. The lowest value is shown

**Table 8. Wilcoxon signed ranks test—task 3.**

| STIMULI PAIR (CODES) | D2-D1 | D3-D1 | D1HD-D1 | D2HD-D1 | D3HD-D1 | D3-D2 | D1HD-D2 | D2HD-D2 |
|---|---|---|---|---|---|---|---|---|
| Z | -2.371[b] | -1.859[b] | -2.366[b] | -2.371[b] | -2.366[b] | -2.356[b] | -.946[b] | -2.271[b] |
| ASYMP. SIG. (2-TAILED) | .018 | .063 | .018 | .018 | .018 | .018 | .344 | .023 |
| STIMULI PAIR (CODES) | D3HD-D2 | D1HD-D3 | D2HD-D3 | D3HD-D3 | D2HD-D1HD | D3HD-D1HD | D3HD-D2HD | |
| Z | -4.280[b] | -.638[c] | -.626[b] | -3.201[b] | -.725[b] | -3.929[b] | -3.557[b] | |
| ASYMP. SIG. (2-TAILED) | .000 | .523 | .531 | .001 | .468 | .000 | .000 | |

[b] Based on positive ranks

[c] Based on negative ranks

Maximum decrease of horizontal distances between symbols with height differentiation applied (D1HD) did not cause significant differences in average solving times compared to D2 and D3 stimuli variants. Significant difference was noted when comparing with the D1 variant (average solving time longer by 01:58, Z = 2.366 p<0.05).

As expected, the D1 variant was the most problematic. A majority of study participants were unable to solve tasks 1 and 3 on this stimuli variant. Yet, we can observe how important the height differentiation of tactile symbols is, when we compare the total number of unsolved tasks 1 and 3 for variants without and with height differentiation applied: 83 and 3 respectively.

The above provides answers to the first and second research questions (RQ1, RQ2). Based on the results presented, we have confirmed the hypothesis that it is possible to reduce the suggested minimum horizontal distances even threefold (from 3 to 1 mm), when applying height differentiation of tactile symbols.

## Participants' feedback

During the study we have also gathered participants' feedback in a form of questionnaires. We wanted to learn their opinions about the quality of the stimuli prepared, their cartographic soundness and if the tactile symbols selected were appropriate. We were especially interested in how the material used for 3D printed maps performed in terms of haptic comfort and also the general level of understanding of the maps presented (Fig 9).

Participants highly appreciated the ease of understanding the maps (26 individuals answered yes or definitely yes) and the comfort of using them (28 individuals answered yes or

**Table 9. Task 3 –ranks.**

| STIMULI PAIR (CODES) | | N | MEAN RANK | SUM OF RANKS |
|---|---|---|---|---|
| D2-D1 | Negative Ranks | 7[a] | 4.00 | 28.00 |
| | Positive Ranks | 0[b] | .00 | .00 |
| | Ties | 0[c] | | |
| | Total | 7 | | |
| D3-D1 | Negative Ranks | 6[d] | 4.17 | 25.00 |
| | Positive Ranks | 1[e] | 3.00 | 3.00 |
| | Ties | 0[f] | | |
| | Total | 7 | | |
| D1HD-D1 | Negative Ranks | 7[g] | 4.00 | 28.00 |
| | Positive Ranks | 0[h] | .00 | .00 |
| | Ties | 0[i] | | |
| | Total | 7 | | |

(*Continued*)

**Table 9.** (Continued)

| STIMULI PAIR (CODES) | | N | MEAN RANK | SUM OF RANKS |
|---|---|---|---|---|
| D2HD-D1 | Negative Ranks | 7[j] | 4.00 | 28.00 |
| | Positive Ranks | 0[k] | .00 | .00 |
| | Ties | 0[l] | | |
| | Total | 7 | | |
| D3HD-D1 | Negative Ranks | 7[m] | 4.00 | 28.00 |
| | Positive Ranks | 0[n] | .00 | .00 |
| | Ties | 0[o] | | |
| | Total | 7 | | |
| D3-D2 | Negative Ranks | 20[p] | 14.35 | 287.00 |
| | Positive Ranks | 7[q] | 13.00 | 91.00 |
| | Ties | 3[r] | | |
| | Total | 30 | | |
| D1HD-D2 | Negative Ranks | 18[s] | 15.47 | 278.50 |
| | Positive Ranks | 12[t] | 15.54 | 186.50 |
| | Ties | 0[u] | | |
| | Total | 30 | | |
| D2HD-D2 | Negative Ranks | 21[v] | 15.36 | 322.50 |
| | Positive Ranks | 8[w] | 14.06 | 112.50 |
| | Ties | 0[x] | | |
| | Total | 29 | | |
| D3HD-D2 | Negative Ranks | 28[y] | 15.73 | 440.50 |
| | Positive Ranks | 2[z] | 12.25 | 24.50 |
| | Ties | 0[aa] | | |
| | Total | 30 | | |
| D1HD-D3 | Negative Ranks | 14[ab] | 13.43 | 188.00 |
| | Positive Ranks | 15[ac] | 16.47 | 247.00 |
| | Ties | 1[ad] | | |
| | Total | 30 | | |
| D2HD-D3 | Negative Ranks | 18[ae] | 12.81 | 230.50 |
| | Positive Ranks | 10[af] | 17.55 | 175.50 |
| | Ties | 1[ag] | | |
| | Total | 29 | | |
| D3HD-D3 | Negative Ranks | 22[ah] | 17.64 | 388.00 |
| | Positive Ranks | 8[ai] | 9.63 | 77.00 |
| | Ties | 0[aj] | | |
| | Total | 30 | | |
| D2HD-D1HD | Negative Ranks | 17[ak] | 14.76 | 251.00 |
| | Positive Ranks | 12[al] | 15.33 | 184.00 |
| | Ties | 0[am] | | |
| | Total | 29 | | |
| D3HD-D1HD | Negative Ranks | 24[an] | 14.69 | 352.50 |
| | Positive Ranks | 3[ao] | 8.50 | 25.50 |
| | Ties | 3[ap] | | |
| | Total | 30 | | |
| D3HD-D2HD | Negative Ranks | 23[aq] | 13.72 | 315.50 |
| | Positive Ranks | 3[ar] | 11.83 | 35.50 |

(*Continued*)

**Table 9.** (Continued)

| STIMULI PAIR (CODES) | | N | MEAN RANK | SUM OF RANKS |
|---|---|---|---|---|
| | Ties | 3[as] | | |
| | Total | 29 | | |

**a.** D2 < D1
**b.** D2 > D1
**c.** D2 = D1
**d.** D3 < D1
**e.** D3 > D1
**f.** D3 = D1
**g.** D1HD < D1
**h.** D1HD > D1
**i.** D1HD = D1
**j.** D2HD < D1
**k.** D2HD > D1
**l.** D2HD = D1
**m.** D3HD < D1
**n.** D3HD > D1
**o.** D3HD = D1
**p.** D3 < D2
**q.** D3 > D2
**r.** D3 = D2
**s.** D1HD < D2
**t.** D1HD > D2
**u.** D1HD = D2
**v.** D2HD < D2
**w.** D2HD > D2
**x.** D2HD = D2
**y.** D3HD < D2
**z.** D3HD > D2
**aa.** D3HD = D2
**ab.** D1HD < D3
**ac.** D1HD > D3
**ad.** D1HD = D3
**ae.** D2HD < D3
**af.** D2HD > D3
**ag.** D2HD = D3
**ah.** D3HD < D3
**ai.** D3HD > D3
**aj.** D3HD = D3
**ak.** D2HD < D1HD
**al.** D2HD > D1HD
**am.** D2HD = D1HD
**an.** D3HD < D1HD
**ao.** D3HD > D1HD
**ap.** D3HD = D1HD
**aq.** D3HD < D2HD
**ar.** D3HD > D2HD
**as.** D3HD = D2HD

**Table 10. Geometric measures of line and area symbols on tactile stimuli.**

| Stimuli variants | | D3/D3HD | D2/D2HD | D1/D1HD |
|---|---|---|---|---|
| **Line symbols** | Total length [mm] | 1935.9 | 1981.3 | 2008.9 |
| | Increase | 0.00% | 2.34% | 3.77% |
| *Area symbols* | Total length [mm] | 1933.0 | 2021.1 | 2128.0 |
| | Increase | 0.00% | 4.56% | 10.1% |
| **Sum of lengths [mm]** | | 3868.9 | 4002.4 | 4136.9 |
| *Average increase* | | 0.00% | 3.45% | 6.93% |

definitely yes). This assessments did not depend on the level of experience with tactile maps (understanding and level of experience: Kruskal-Wallis test $\chi2(3) = 3.341$, $p = 0.188$; comfort and level of experience: Kruskal-Wallis test $\chi2(3) = 3.245$, $p = 0.197$), which suggests that the assessment is related to the map properties.

Study participants made a number of suggestions on how to modify tactile stimuli for better distinguishability and legibility. All of the feedback presented in this section has been mentioned by at least 3 participants of the study.

We asked the participants whether some symbols were too similar or too close to each other that made them hard to perceive. In general, many participants were confusing circle with triangle and square. One of the suggestions that could prevent misidentification was to use both types of point symbols within one map: full-infill and outlines only. This would result in different roughness when touching a symbol. Besides, it turns out that rounding tactile symbols can cause confusion when it comes to distinguishing similar shapes, e.g. square from circle. We have applied rounding to raise the tactile comfort. Thus, it is a matter of trade-off between legibility and comfort.

Another interesting suggestion was to use isosceles triangles instead of equilateral ones to differ them from circles even more. More generally, the point symbols should be bigger according to the participants.

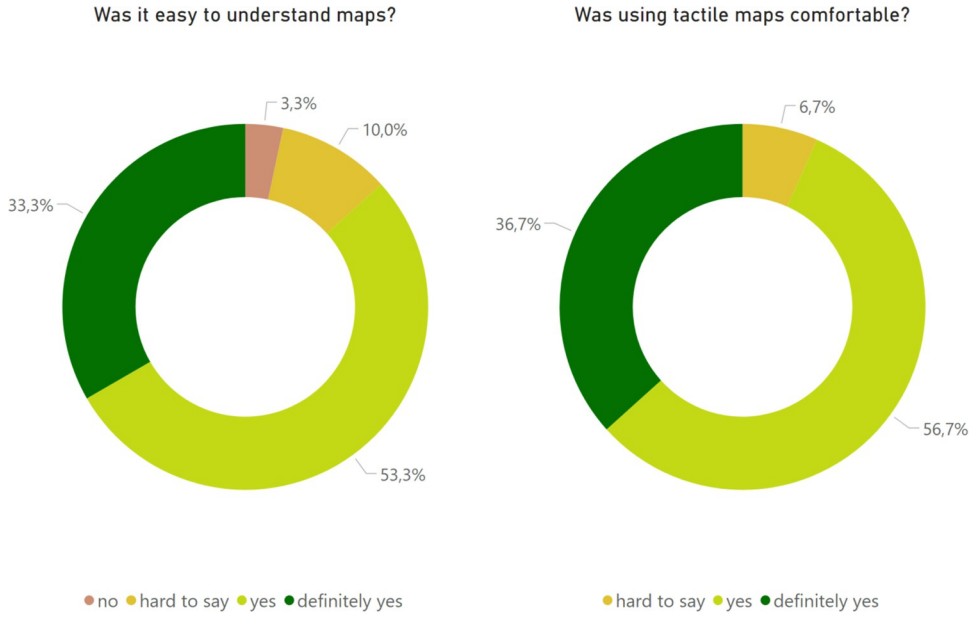

**Fig 9. Results of the post-study questionnaire.**

Another very common comment was that sometimes it was hard to find circles (task 1) or paths (task 3) that were "hidden" within textures (area symbols). Participants who realized that in some stimuli variants height differentiation of tactile symbols was applied, pointed out the importance of this design feature. Some participants, even without noticing this modification, said that it would be very helpful to put point symbols above the textures to identify them more easily.

Some participants indicated too similar pairs of textures (e.g. zig-zag and line patterns) but the comments were very diverse and we were not able to determine, which pattern was the most confusing one.

During the study, we have also asked participants about their personal opinions and comments about the study and how to improve the future tactile maps. We have selected some of the most popular comments:

- Many of the participants pointed out that the symbols used on the stimuli have different meanings on maps that they know. For example, one of the textures used on the stimuli is commonly used for depicting water bodies on Polish tactile maps. Others expressed the need for tactile symbols standardization more explicitly.

- Even though we have applied height differentiation on selected stimuli variants, some participants said that it would be a good idea to differentiate them even more.

- There was no consistency in terms of proper minimum distance between symbols on tactile maps. Based on the comments gathered and our observations during the study, the higher distances sometimes caused confusion when tracking a path (task 3)–"is it still the same path or a new one?", whereas too low distances caused the participants to select wrong paths at the crossroads. Perhaps, a 2 mm distance would be the most optimal choice?

## Information value gain

Using the 3D modelling software, we have calculated the lengths of particular tactile symbol types on each of the stimuli variants' vector drawings. In case of line symbols, the lengths of axes were evaluated, whereas for area symbols, we have measured their outline lengths (Fig 10). Point symbols were not taken into consideration in this study as their number did not vary across stimuli variants.

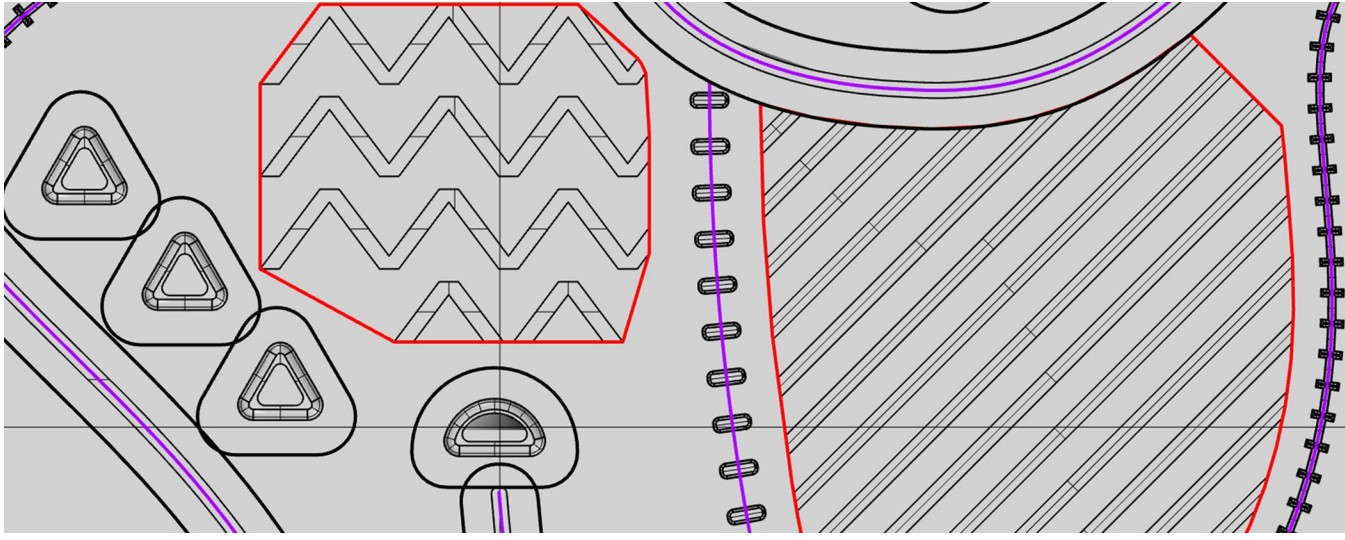

**Fig 10. Vectors used for calculation of information value.** Lines' axes are marked in purple, area symbols' outlines in red.

The measures taken have been used for calculation of potential information value gain thanks to the reduction of minimum horizontal distances between symbols (Table 10).

The potential increase of information value reaches over 10% in terms of area symbols for minimum horizontal distance reduction from 3 to 1 mm. If we average the two evaluated symbol categories, we will get approximately a 3.5% and 6.9% increase when reducing the minimum distances to 2 and 1 mm respectively. This is a significant increase if we consider the costs of tactile map production and how complex the process of tactile map generalization is. This analysis provides an answer to the third research question (RQ3).

The real information gain could be even higher as the additional haptic variable of height can introduce new spatial information, e.g. the higher the symbol representing a city, the more population this city has, as it was previously described by Wabiński et al. [11].

## Discussion

The results of our study further extend the conclusions described by Nolan and Morris [19]. In their study of seeking optimum tactile symbol design for maximum legibility, one of the tasks was to follow the path denoted by a dotted line on photo engraved tactile maps. Their results proved the application of height differences to be a significant factor in time required to follow the path, whereas minimum horizontal distances between particular symbols were not significant. Thus, we can conclude that the findings are not dependent on the production method used.

Based on the participants' feedback we can assume that the issues raised in previous studies regarding difficulties in access to high quality tactile maps [37] still exist. PVI have no experience in working with tactile maps as they are often not even aware of the existence of such materials in public spaces. For this reason not only the issues with production of new tactile maps should be emphasized but also the ways to inform about the existing ones.

Until today, there was not much research comparing various tactile maps production methods. Previous studies did not agree on the best production approaches [38, 39]. But using 3D printing for tactile map production looks promising. 3D printed maps tend to be rough in touch but as it was stated in the past [19, 40, 41] as well as in our qualitative analysis, PVI prefer this kind of tactile stimuli.

Our long-term goal is to convince users and producers of tactile maps that 3D printing can be successfully used for generation of cheap, legible, and unique map sheets. According to Leonard & Newman [42], researchers should look for standardized procedures that can be applied locally and at relatively low cost and without expensive equipment in terms of tactile aids production. This was an issue back then but even now, more than 50 years later, many PVI lack tactile aids. This is also true for developed countries [25, 43].

In order to quickly generate cheap tactile maps, a repeatable process of their development is necessary. This requires standardization, including strictly defined parameters of symbol design and map editing. They make it possible to automate and speed up the entire process of map generation. Previous studies have not developed unequivocal parameters, but rather recommendations. These recommendations usually do not inform, what printing technique should be used along with them. Therefore, they are not enough to standardize the process of tactile maps development.

3D printing makes it possible to use an additional tactile variable–height, that is currently rarely used. Using this additional variable, it was possible to reduce the distances between the symbols in our study. In this context, the existing recommendations regarding the designed symbols and map redaction should be redefined. The parameters for 3D printing technique

were defined in our study and could be reused in the future. They can be the starting point for standardization and automation of tactile maps development process.

Increasing the readability of maps using height differentiation is not dependent on personal factors–no significant differences were found in average solving times of both congenitally/ early and adventitiously blind or those with different levels of Braille reading skills and experience with tactile maps.

Using 3D printing as a method for production opens many possibilities in terms of fast and accurate development of not only tactile maps, but educational tactile materials in general. As it turned out, the vast majority of our study participants (over 90%) indicated their satisfaction when using 3D printed tactile stimuli. Most of them found working with 3D printed tactile maps pleasant and not tiring (even in the face of relatively long study sessions). At the same time, many of them were very intrigued by the possibility of printing similar maps on demand in a fast and cheap way using such printing method. Educators across the world should consider implementing 3D printing in their labs as a method for tactile materials preparation, especially in the face of the rapid development of this technology that leads to significant cost reduction of printing equipment and materials.

## Conclusions

This paper's aim was to systematically assess our hypothesis that using height differentiation of symbols on tactile maps facilitates their reading and allows reduction of minimum distances between particular symbols. Our results confirmed this hypothesis.

Thanks to the qualitative analysis conducted during the study sessions, we have learned a lot about expectations of PVI in terms of tactile maps design. We consider this a significant step towards more efficient and cheaper tactile map production and possibly, the automation of this process, along with standardization of tactile symbols on maps.

In this study we wanted primarily to compare stimuli variants: D3 and D1HD as they represented our hypothesis in the clearest manner. D3 is the variant prepared according to the existing good practices. We wanted to prove that applying height differentiation may allow reduction of the minimum distances between tactile symbols (D1HD). When looking at the quantitative results of our study, we can see that average solving times for the D1HD variant were either definitely lower (task 1) or comparable (tasks 2 and 3). Although the differences in solving times between these two variants were not statistically significant, D1HD was more efficient when it came to identifying point and area symbols as well as–to a lower extent–line symbols. Thus, we can conclude that it is possible to reduce the suggested minimum horizontal distances between symbols on tactile maps by applying height differentiation and at the same time, gain additional space that may be used to put more relevant information or simply increase map's legibility.

We believe that our study has once again shown the importance of tactile maps design standardization. Some study participants were disoriented by some of the symbols used, even though they were carefully selected.

In future research we would like to further evaluate possibilities of tactile maps information value improvement by differentiating heights of tactile symbols within one geometry type. Besides, we want to focus on the issue of tactile symbols standardization. The first steps have been taken and we would like to involve the practitioners across the globe to share their experiences in this area.

Besides, we plan to apply the knowledge gained during this study to form parameters that will control the process of automatic or semi-automatic tactile map generalization out of digital spatial data publicly available thanks to initiatives, such as the INSPIRE directive [44].

Besides, we would like to analyse map reading techniques and participants' behaviour based on the video material recorded during the study sessions to learn about the most efficient ways of tactile maps scanning.

This study has left us with a number of valuable suggestions on how to improve tactile maps that should be considered in the processes of tactile map development. Besides, they ought to form a part of potential official tactile maps design guidelines.

However, the needs of every PVI differ. Particular symbols were confused by some participants, whereas others had no troubles in distinguishing them. Some participants preferred the legend to be separate from the map, whereas others would like it to be an integral part of the map. All that allows us to confirm past conclusions that preparing tactile map design standards suitable for all PVI is a real challenge.

## Supporting information

**S1 File. Research protocol.** Formal description of the testing phase.
(DOCX)

## Acknowledgments

We would like to thank professor Amy Lobben for her invaluable support in the study design and validation.

## Author Contributions

**Conceptualization:** Jakub Wabiński, Albina Mościcka.

**Data curation:** Jakub Wabiński, Emilia Śmiechowska-Petrovskij.

**Formal analysis:** Jakub Wabiński, Emilia Śmiechowska-Petrovskij.

**Investigation:** Jakub Wabiński, Emilia Śmiechowska-Petrovskij.

**Methodology:** Jakub Wabiński.

**Supervision:** Albina Mościcka.

**Validation:** Emilia Śmiechowska-Petrovskij.

**Writing – original draft:** Jakub Wabiński, Emilia Śmiechowska-Petrovskij.

**Writing – review & editing:** Albina Mościcka.

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
