## [Decision Letter · Decision Letter 0]

17 Jan 2022

PONE-D-21-29348Applying height differentiation of tactile signs to reduce the minimum horizontal distances between them on tactile mapsPLOS ONE

Dear Dr. Jakub Wabiński,

Thank you for submitting your manuscript to PLOS ONE. After careful consideration, we feel that it has merit but does not fully meet PLOS ONE’s publication criteria as it currently stands. Therefore, we invite you to submit a revised version of the manuscript that addresses the points raised during the review process.

The points made by one of the two reviewers are relate to the substantive parts of analysis. Please note that your revision will undergo another round of review before a final decision is rendered.

We look forward to receiving your revised manuscript.

Kind regards,

Maki Sakamoto, Ph.D

Academic Editor

PLOS ONE

Journal Requirements: 

2. We note that Figure 6 includes an image of a participant.

If you are unable to obtain consent from the subject of the photograph, you will need to remove the figure and any other textual identifying information or case descriptions for this individual

Reviewers' comments:

Reviewer's Responses to Questions

**Comments to the Author**

1. Is the manuscript technically sound, and do the data support the conclusions?

Reviewer #1: Yes

Reviewer #2: Yes

2. Has the statistical analysis been performed appropriately and rigorously? 

Reviewer #1: Yes

Reviewer #2: No

3. Have the authors made all data underlying the findings in their manuscript fully available?

Reviewer #1: Yes

Reviewer #2: Yes

4. Is the manuscript presented in an intelligible fashion and written in standard English?

Reviewer #1: Yes

Reviewer #2: Yes

5. Review Comments to the Author

Reviewer #1: In this manuscript, the understanding of the tactile map was improved by the height differential signs.

In addition, it was suggested that the height differences can reduce the recommended minimum horizontal distance between signs on the tactile map, and provide more relevant information or simply make the map easier to read.

I think that this manuscript is acceptable for publication after minor revision.

1) Three tasks are performed in this study. but, the task numbers are not shown in "Human subject testing". I suggest to you that you write as follows in lines 237 to 242: "Task 1. Locate 5 point symbols ...", "Task 2. Locate 5 area symbols ..." and "Task 3. Follow the path ...".

2) The table numbers quoted in the document are incorrect as follows:

Line 351: “table 4-6"  "table 3-5".

Line 392: "Table 7"  "Table 6".

Line 402: “tables 8-10”  “table 7-9”.

Line 501: “Table 3”  “Table 10”.

Reviewer #2: The transformation of digital map into tactile form is an important issue in efforts to compensate information for the visually impaired, and creating a tactile map will require a certain amount of standardization. This paper tested weather cartographic signs on tactile maps to different heights can reduce the minimum horizontal distances between them. This test will lead to efficient production of tactile maps. However, there are a few concerns about the testing.

1. The authors used 6 different stimuli that have different horizontal distances between tactile signs (D1, D1HD, D2, D2HD, D3, D3HD). It is better to show what all the stimuli look like in the Figure, and to propose hypothesis in advance about what we can learn by comparing what with what regarding three Research Questions proposed by the authors.

2. It seems predictable that the performance would be better with height difference, and the stimuli used in this experiment seem arbitrarily chosen. It is not clear how they will contribute to standardization of tactile map directly.

3. As for Performance results, from my understanding, the comparisons between D1-D1HD, D2-D2HD, and D3-D3HD are essential. Why are the authors making statistical comparisons even for unrelated combinations? The statistical descriptions seem to have become cumbersome.

4. Is there any difference in performance between people who are congenitally and adventitious blind? If so, the analysis is performed separately.

6. PLOS authors have the option to publish the peer review history of their article (what does this mean?). If published, this will include your full peer review and any attached files.

Reviewer #1: No

Reviewer #2: No

---

## [Author Response · Author response to Decision Letter 0]

26 Jan 2022

A separate file marked as 'Response to Reviewers' has been uploaded along with resubmission. Please refer to this file.

---

## [Decision Letter · Decision Letter 1]

14 Feb 2022

Applying height differentiation of tactile symbols to reduce the minimum horizontal distances between them on tactile maps

PONE-D-21-29348R1

Dear Dr. Wabiński,

We’re pleased to inform you that your manuscript has been judged scientifically suitable for publication and will be formally accepted for publication once it meets all outstanding technical requirements.

Kind regards,

Maki Sakamoto, Ph.D

Academic Editor

PLOS ONE

Additional Editor Comments (optional):

Reviewers' comments:

Reviewer's Responses to Questions

**Comments to the Author**

1. If the authors have adequately addressed your comments raised in a previous round of review and you feel that this manuscript is now acceptable for publication, you may indicate that here to bypass the “Comments to the Author” section, enter your conflict of interest statement in the “Confidential to Editor” section, and submit your "Accept" recommendation.

Reviewer #1: All comments have been addressed

Reviewer #2: All comments have been addressed

2. Is the manuscript technically sound, and do the data support the conclusions?

Reviewer #1: Yes

Reviewer #2: Yes

3. Has the statistical analysis been performed appropriately and rigorously? 

Reviewer #1: Yes

Reviewer #2: Yes

4. Have the authors made all data underlying the findings in their manuscript fully available?

Reviewer #1: Yes

Reviewer #2: (No Response)

5. Is the manuscript presented in an intelligible fashion and written in standard English?

Reviewer #1: Yes

Reviewer #2: Yes

6. Review Comments to the Author

Reviewer #1: (No Response)

Reviewer #2: After the revision, images of the stimuli are presented and the relationship between the hypothesis and the experiment is shown to make the argument easier to understand.

7. PLOS authors have the option to publish the peer review history of their article (what does this mean?). If published, this will include your full peer review and any attached files.

Reviewer #1: No

Reviewer #2: No

---

## [Editor Report · Acceptance letter]

18 Feb 2022

PONE-D-21-29348R1 

Applying height differentiation of tactile symbols to reduce the minimum horizontal distances between them on tactile maps 

Dear Dr. Wabiński:

I'm pleased to inform you that your manuscript has been deemed suitable for publication in PLOS ONE. Congratulations! Your manuscript is now with our production department. 

Kind regards, 

on behalf of

Dr. Maki Sakamoto 

Academic Editor

PLOS ONE